# Sustainability benefits of transitioning from current diets to plant-based alternatives or whole-food diets in Sweden

Anne Charlotte Bunge [1] ✉, Rachel Mazac [1,2,3], Michael Clark [4,5,6], Amanda Wood [1] & Line Gordon [1]

Plant-based alternatives (PBAs) are increasingly becoming part of diets. Here, we investigate the environmental, nutritional, and economic implications of replacing animal-source foods (ASFs) with PBAs or whole foods (WFs) in the Swedish diet. Utilising two functional units (mass and energy), we model vegan, vegetarian, and flexitarian scenarios, each based on PBAs or WFs. Our results demonstrate that PBA-rich diets substantially reduce greenhouse gas emissions (30–52%), land use (20–45%), and freshwater use (14–27%), with the vegan diet showing the highest reduction potential. We observe comparable environmental benefits when ASFs are replaced with WFs, underscoring the need to reduce ASF consumption. PBA scenarios meet most Nordic Nutrition Recommendations, except for vitamin B12, vitamin D and selenium, while enhancing iron, magnesium, folate, and fibre supply and decreasing saturated fat. Daily food expenditure slightly increases in the PBA scenarios (3–5%) and decreases in the WF scenarios (4–17%), with PBA diets being 10–20% more expensive than WF diets. Here we show, that replacing ASFs with PBAs can reduce the environmental impact of current Swedish diets while meeting most nutritional recommendations, but slightly increases food expenditure. We recommend prioritising ASF reduction and diversifying WFs and healthier PBAs to accommodate diverse consumer preferences during dietary transitions.

Large-scale dietary changes are urgently needed to achieve sustainable and resilient food systems within the planetary boundaries while ensuring nutritious diets for a growing world population[1–3]. A range of modelling studies have found that, globally, the shift to more plant-based diets would mitigate the environmental pressures from food systems while promoting public health[4,5]. These studies often analyse dietary scenarios, where animal-source foods (ASFs) are replaced by whole foods (WFs) such as legumes. However, consumer studies revealed that

key barriers to moving away from ASFs and adopting such diets based on whole foods are the taste, convenience, and cultural values of eating ASFs[6].

To overcome these barriers to decreasing ASF consumption and consuming more sustainable diets, an alternative is to replace conventional ASFs such as meat, dairy, and seafood with novel plant-based alternatives (PBAs). Contrary to WFs, PBAs are processed foods that aim to mimic the structure, texture, and sensorial properties of the ASF they intend to replace[7]. This means they can support transitions

[1]Stockholm Resilience Centre, Stockholm University, Stockholm, Sweden. [2]Helsinki Institute of Sustainability Sciences (HELSUS), University of Helsinki, Helsinki, Finland. [3]Department of Agricultural Sciences, Faculty of Agriculture and Forestry, University of Helsinki, Helsinki, Finland. [4]Smith School of Enterprise and the Environment, Oxford, UK. [5]Department of Biology, University of Oxford, Oxford, UK. [6]Oxford Martin School, University of Oxford, Oxford, UK. ✉e-mail: annecharlotte.bunge@su.se

towards more plant-based diets among consumers who strive to reduce their ASF intake but are not willing to compromise convenience or desirable sensory attributes[8]. Hence, while PBAs are often based on WFs, such as legumes, they differ in the way they are processed and in their sensory profiles.

PBAs are promoted for their potential to mitigate dietary environmental impacts, animal welfare, nutritional and health, and food safety concerns that are linked to ASFs[9]. A range of comparative life cycle assessment (LCA) studies revealed that PBAs are more environmentally sustainable compared to their ASF equivalents[10–12], but less compared to unprocessed WFs[9]. In addition, there is uncertainty about the nutritional impacts and long-term health consequences of consuming PBAs, with some research questioning the benefits of PBAs[13] while others found that PBAs provide more nutritious alternatives to processed ASFs[8,14].

The number of PBAs available in retail stores has steadily increased in recent years[15]. In the European Union, consumption increased 49% between the years 2018 and 2020, with the highest growth reported in Germany (226%)[16]. Hence it can be assumed that integrating PBAs into diets reflects an emerging dietary behaviour in several countries. More empirical evidence is therefore imperative to understand the broad sustainability implications of scaling up PBA consumption, including environmental, nutritional, and socioeconomic aspects.

As pathways for more sustainable diets are highly context-specific- depending on the national burden of diseases, environmental challenges related to respective food systems and cultural traditions[17]—such studies would benefit from being situated in specific cultural contexts. Moreover, most previous analyses on PBAs focused on the product level or assessed only subsets of PBAs in dietary scenario studies, primarily meat replacements[18–20] with only a few taking a whole dietary perspective to understand the role of PBAs in sustainable diets[21]. Additionally, these studies often compared PBAs only with ASFs but not with WFs.

Here, we broaden the focus of PBAs to assess meat, seafood, dairy (including cheese), and snack replacements that mimic their respective ASF and focus on products that are already retail available. We use Sweden as a case study for two reasons. First, the per capita consumption of ASFs[22] is high, with a high environmental[23] and epidemiological burden from current average Swedish diets[24]. Second, the market of available PBAs is growing[25] with some PBAs already being included in governmental communication on sustainable diet strategies[26].

In this study, we use country-specific data to conduct a multi-indicator analysis by modelling the environmental, nutritional, and economic impacts of current diets and flexitarian, vegetarian, and vegan dietary scenarios that replace ASFs with either PBAs or WFs. Overall, our findings show that these dietary choices can reduce the environmental impact of current Swedish diets while being cost-competitive and meeting most nutritional recommendations.

## Results

### Nutritional adequacy of the dietary scenarios
The nutrient profile of the six scenarios improved in several ways relative to the current Swedish diet (BAU) and met most Nordic Nutrition Recommendations with some exceptions (Fig. 1, Supplementary Tables 3 and 4, and Supplementary Data 1). Nutrient content increased for iron (15–47%), fibre (36–163%), folate (13–160%), magnesium (13–147%) and polyunsaturated fats (26–92%) and decreased for saturated fats (13–38%) in all the six scenario diets compared to the BAU. Protein and zinc intake met and exceeded dietary recommendations in all six scenario diets with no protein or zinc deficiency risk. However, the protein and zinc content decreased in the six scenario diets compared to the BAU (6–22% and 20–64%, respectively), with the

lowest content among the vegan WFs (VGNWHOLE) and vegan PBAs (VGNPBA) scenarios. All scenarios met the recommendation for calcium intake, except the VGNWHOLE and flexitarian WF (FLXwhole) diet. B12 decreased to less than recommended levels among the VGNWHOLE and VGNPBA scenarios. All scenarios including the BAU met the iron, magnesium, phosphorus, vitamin E and C recommendations, but none met those for selenium. Vitamin D content decreased in all scenarios and only VGNPBA and flexitarian PBAs (FLXPBA) remained adequate intake, which is attributable to the vitamin D fortification of included PBAs. The sodium content was above recommendations in the BAU and increased in the PBA scenarios (11–24%) while it decreased in the WF scenarios to the recommendations.

### Environmental impact
Shifting to more plant-based PBA or WF diets was estimated to lead to reductions in greenhouse gas emissions (GHGe) and land use (LU) for both functional units (Fig. 2; Supplementary Fig. 5). Adopting vegan diets revealed the highest estimated reduction for GHGe (52% VGNPBA, 56% VGNWHOLE) and LU (44% VGNPBA, 32% VGNWHOLE), followed by vegetarian and flexitarian diets. The lowest but moderate estimated reduction potential for GHGe (30%, 20%) and LU (22%, 27%) was revealed for the FLXPBA and FLXWHOLE scenarios, respectively. Moderate freshwater use (WU) reduction (14–26%) was revealed in all scenarios compared to the BAU diet, except for FLXWHOLE and VGNWHOLE where it remained at the same level, which can be attributed to increased fruit and vegetable consumption in these scenarios.

In current diets, ASF consumption contributed the most to GHGe (75%) even though they accounted for 37% of total energy. This was predominantly from meat products (43% total GHGe), which were also the largest contributor to impacts on LU (67%) and WU (38%). When all ASFs were replaced by PBAs in the VGNPBA scenario, total GHGe were reduced by 50%, and PBAs contributed 37% of total GHGe. The main contribution to GHGe in the VGNPBA scenario was from PBA Snacks (26%) mostly attributable to dark chocolate, which has a high GHG impact per unit of food produced. Detailed results displaying the proportional contributions of each food group to the total environmental impacts of each dietary scenario are provided in the Supplementary Data.

### Daily food expenditure
We estimated that transitions from current diets to more plant-based diets resulted in small to moderate changes in daily food expenditure (Fig. 3). The average cost per day of the BAU diet in 2022 amounted to SEK 82 (Q25th-Q75th: SEK 66-105). We estimated small increases in diet costs when shifting towards PBAs diets (3-5%), except for VGTPBA where costs remained similar, while shifting towards WFs diets decreased estimated daily food expenditure (4-17%).

The greatest cost reduction potential was revealed for the VGTWHOLE Scenario (17%), where fruits and vegetables accounted for the largest expense (23%), followed by legumes and nuts (20%). In the BAU diet, ASFs accounted for the greatest proportion of expenses (50%) with 28% related to meat products. Exchanging ASFs with their respective PBAs led to similar price values. For example, meats and PBA meats had similar median price values as well as dairy and PBA dairy, but PBA seafood was priced higher (20%) than the median price for seafood. Expenses per dietary scenario and food category on a mass and energy basis stratified for different quartiles are provided in the Supplementary Data.

### Uncertainty analysis
Our findings were similar when using the mass-based and energy-based functional units (Supplementary Results). In the nutrient analysis, using the energy-based functional unit slightly reduced the differences between the current diet and the diet scenarios. For example,

a lower amount of dietary protein was revealed in all replacement scenarios (63–84 g compared to 74–90 g using a mass-based unit) but still remained above recommendations. The results were also similar for the environmental analysis, where estimated GHGe were 7–23% higher using the energy-based functional unit (2.6–3.7 kgCO2eq compared to 2.1–3.4 kgCO2eq), except for VGTPBA, but still remained 22–45% below the BAU. The slight environmental impact increase in the WHOLE scenarios is mainly attributable to the higher share of legumes and nuts when replaced by unit energy. In the PBA scenarios, the environmental impact of PB dairy increased while it decreased for PB meat. Daily food expenditure slightly increased using the energy-based unit, except for VGTPBA, with the vegetarian scenarios revealing the lowest cost expenditure.

In the second uncertainty analysis, we assessed the potential variation in nutritional, environmental, and diet cost impacts when accounting for differences within food types (Supplementary Results). This is indicative of how a conscious consumer might act in a retail store, for instance preferentially purchasing cheaper, more environmentally sustainable, or nutritious foods. When accounting for this variation we found a high variation in the protein and iron content of meats and PB meat (Supplemental Fig. 4). We revealed low variances in

the environmental performance of legumes and PBAs. Our findings suggest large price ranges for several food items and food groups, such as seafood (+70% in Q75; +150% in Q100). The high price variation revealed that the BAU diet would be the most cost-expensive if maximum price values were purchased, using both functional units.

## Discussion

Adopting more plant-based diets based on either PBAs or WFs revealed several nutritional implications such as higher fibre, folate, and lower saturated fat content which could provide nutritional benefits for the Swedish population, where the majority currently does not meet dietary recommendations[27]. The sodium content increased in the PBA scenarios compared to the BAU (11–24%) adding to the concern that PBAs have too high levels of added salt[13,28], while it decreased in the WF scenarios. Too high sodium intake is a major Public Health concern in the European region[29]. Some micronutrient intakes would be further reduced from current levels, such as vitamin D. However, vitamin D is considered a critical nutrient of concern due to the northern latitudes, daily supplementation is already recommended in the Nordic Nutrition Recommendations[30]. Zinc intake was well above the recommended intake, despite being lower than the BAU in all scenarios. Many PBAs

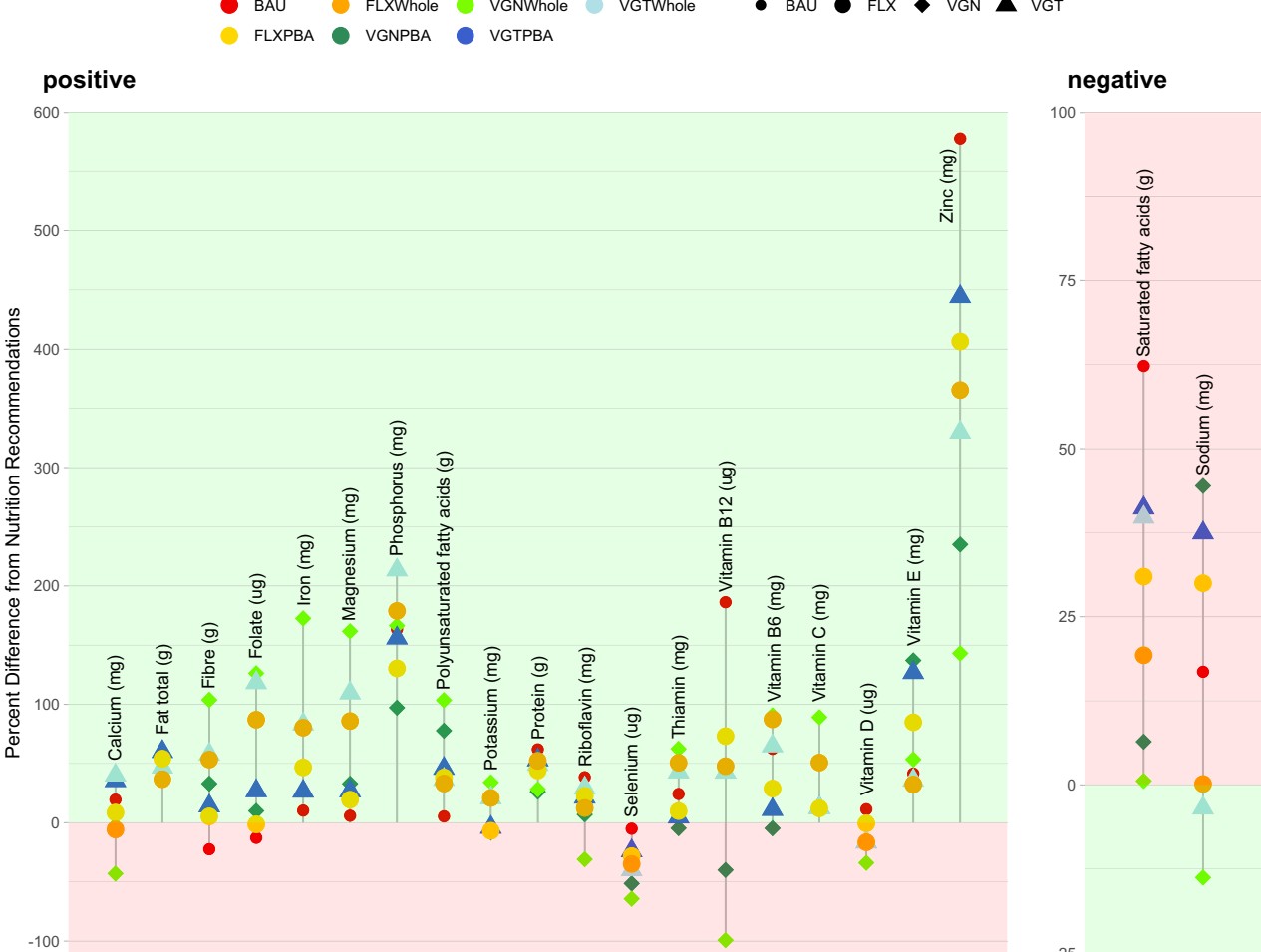

**Fig. 1 | Nutritional performance of the current diet and alternative dietary scenarios in alignment with the Recommended Dietary Allowances provided by the Nordic Nutrition Recommendations (0 = respective recommended nutrient level).** Green space (above 0 in the positive nutrients, below 0 in the negative) indicates meeting or extending the recommended levels, red space (below 0 in the negative nutrients, above 0 in the positive) indicates not meeting (or exceeding in nutrients to limit) recommended levels. BAU Average Swedish Diet, VGNPBA Vegan Diet, all animal source foods (ASFs) replaced by plant-based alternatives (PBAs), VGTPBA Vegetarian Diet, meat and seafood replaced by respective PBAs, FLXPBA 50% reduction of ASFs replaced by PBAs, VGNWHOLE Vegan Diet, all ASFs replaced by WFs), VGTWHOLE Vegetarian Diet, meat and seafood replaced by WFs, FLXWHOLE 50% reduction of ASFs replaced by WFs. Source data are provided as a Source Data file.

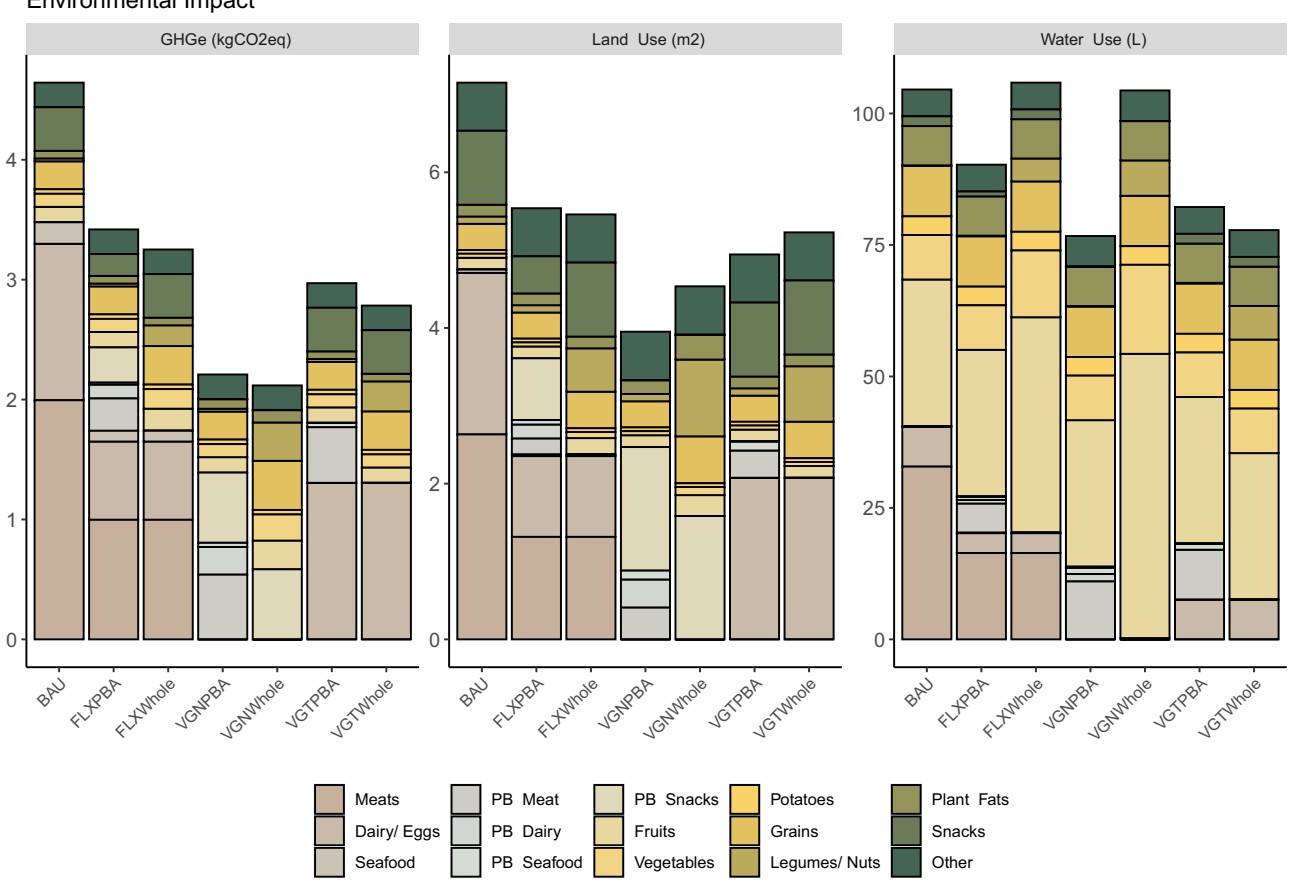

**Fig. 2 | Environmental impact of the current diet and alternative dietary scenarios based on a mass-based functional unit.** Contribution by different food categories to impact categories. The category "other" includes sugar and sugar-based products, salt, soft drinks, coffee (roasted powder) and juice. BAU Average Swedish Diet; VGNPBA Vegan Diet, all animal source foods (ASFs) replaced by plant-based alternatives (PBAs); VGTPBA Vegetarian Diet, meat and seafood replaced by respective PBAs; FLXPBA = 50% reduction of ASFs replaced by PBAs; VGNWHOLE Vegan Diet, all ASFs replaced by whole foods (WFs); VGTWHOLE Vegetarian Diet, meat and seafood replaced by WFs; FLXWHOLE = 50% reduction of ASFs replaced by WFs. The acronym PB in the food category means plant-based. Source data are provided as a Source Data file.

available at Swedish retail are fortified with B12, vitamin D, calcium, and iron to prevent potential micronutrient deficiencies[14,31], which is reflected in our findings that the micronutrient content of the VGNPBA did not largely differ from the BAU diet. Low selenium intake was revealed in all scenarios, including the BAU, which reflects existing evidence that selenium intake is a concern in Sweden[32]. Simultaneously, there is a discussion on the bioavailability of these fortified nutrients in PBAs, requiring more research on the bioavailability of vitamins and minerals as well as the protein quality of PBAs for a better understanding of the nutritional effects of integrating them into diets[14].

Our results suggest that shifting to both PBA and WF vegan diets would reveal the largest environmental impact reduction potential in terms of LU and GHGe, followed by vegetarian and flexitarian diets. In the VGNPBA scenario, the highest share of GHGe and LU resulted from the PB snacks category. Here, we used dark chocolate as a proxy for PB chocolate, which has a larger environmental footprint than milk chocolate due to its higher cocoa content[33]. Choosing PB milk chocolate based on oat drinks would likely have yielded a lower environmental impact but was limited by unavailable LCA data. PB dairy, seafood and meat alternatives revealed significantly lower environmental impacts than ASFs, in line with accumulating evidence from comparative LCA studies[9,10,12].

Economically, we estimated that all assessed scenarios were similar with the BAU diet when comparing median prices. According to our results, shifting towards diets high in PBAs slightly increases expenditure, while shifting towards WFs diets slightly decreases the daily food

expenditure, in line with previous findings concerning high-income countries[20,34]. These results suggest that healthier and more environmentally sustainable diets can be obtained without much alteration of food expenditure costs in Sweden, depending on which end of the price range consumers make their purchases, raising important questions about diet-affordability in general. When compared with data provided by the Swedish Consumer Agency our results revealed a moderate underestimation of daily food expenditure. According to their latest estimation, adults in Sweden will spend between 102-120 SEK per day on food in the year 2023 (~20% more than our results revealed but in line with Q75th)[35]. Several reasons can be used to explain this divergence. First, we extracted food prices from a retailer that is considered relatively low-price[36]. Second, we used market price data from June 2022 and food prices have since been highly affected by inflation[37]. Lastly, since we excluded food waste at household stage in the scenarios, the data is based on consumption data, not purchase data, and therefore already provides an underestimation.

Our analysis advances the literature on the implications of integrating PBAs into diets in several ways. First, while the literature on assessing the sustainability performance of PBAs is growing, previous studies primarily compared PBAs with ASFs on an individual product level. On the diet level, previous modelling studies assessed a subset of PBAs, primarily meat substitutes[18–20], but rarely focused on dairy and seafood alternatives. Here, we not only expand the analysis to a wider variety of PBAs now available in retail and thus include them as new

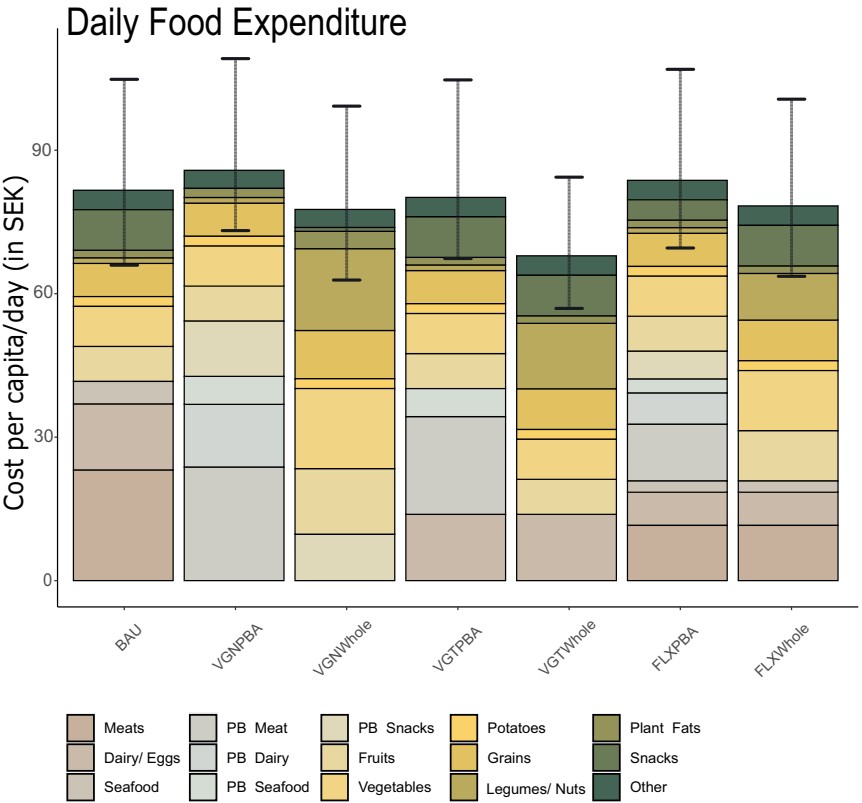

**Fig. 3 | Daily Food Expenditure (SEK per day) in June 2022 by dietary scenario and food group.** 10 SEK = 0,9€/0.96 US$. Bar height corresponds to median values of daily food expenditure. Error bars indicate the lower (25th) and upper (75th) percentile for the daily food expenditure aggregated for all food groups (*n* = price for 940 food items aggregated in *n* = 15 food groups). Other include sugar and sugar-based products, salt, soft drinks, and juice. BAU Average Swedish Diet; VGNPBA Vegan Diet, all animal source foods (ASFs) replaced by plant-based alternatives (PBAs); VGTPBA Vegetarian Diet, meat and seafood replaced by respective PBAs; FLXPBA = 50% reduction of ASFs replaced by PBAs, VGNWHOLE Vegan Diet, all ASFs replaced by whole foods (WFs); VGTWHOLE Vegetarian Diet, meat& seafood replaced by WFs; FLXWHOLE = 50% reduction of ASFs replaced by WFs. The acronym PB in the food category means plant-based. Source data are provided as a Source Data file.

food categories in dietary modelling studies but also compare them against WFs as alternative replacements.

Second, we used national nutritional composition data and assessed the nutritional performance of the respective scenarios in relation to the Nordic Nutrition Recommendations. This comparison to established nutrient criteria, rather than relative to other diets, reduces the risk of misinterpretation. For example, our results highlight that even if plant-based diets based on PBAs or WFs provided less protein content, they still met and exceeded dietary recommendations, supporting the fact that European diets are very high in protein content already[38].

Third, we built six scenarios using two different functional units, mass basis and energy content basis following previous studies that asserted choosing the functional unit has a decisive impact on the sustainability performance of food items[39,40]. Both methodological assumptions have their limitations, hence assessing scenarios built on both functional units disentangles the nuances of identifying sustainable diets.

Lastly, by using country-specific data (e. g. retail price data) we were able to provide a more geographically and culturally explicit case study. Such localisation may prove useful in both Swedish and Nordic food system transformation planning and implementation[41]. However, while we here provide a case study for the Swedish food supply chain, our multi-indicator analysis has useful implications for a broader context by providing a methodological template that can be adjusted to other countries given accessible national data.

Several limitations apply to our study as well, mainly attributable to data availability. First, consumption data for PBAs and tofu was not provided in the Swedish national database and determining the intake of legumes proved difficult as they were not listed separately but included in vegetable food categories. Hence and in line with the available dietary intake data, we set the consumption data of PBAs, legumes, and soy foods to zero in the BAU diet, which does not reflect consumers already including PBAs or legumes in their diet. Including legumes in the BAU diet would have slightly altered the environmental, economic, and nutritional performance, such as a potential increase in fibre intake. Second, focusing on the potential implications of the bioavailability of fortified nutrients in currently retailed PBAs and addressing the effects of processing was outside the scope of this research and is detailed elsewhere[14,31,42,43]. We here used nutritional content data for PBAs that are currently available at retail. However, PBAs as processed foods can be reformulated to improve their healthiness[44], with particular emphasis on reducing sodium content, which our findings revealed is currently too high in PBAs, and promoting nutrients of concern in the Swedish population. In Sweden among other countries, ongoing research focuses on developing next-generation PBAs that will be healthier, tastier, more sustainable, and produced locally from Swedish raw materials[45]. For example, by using fermentation practices to improve the bioavailability of nutrients in PBAs[46]. Contrary, ultra-processing has been linked to various adverse long-term health outcomes, highlighting the necessity for further research into the health implications of diets rich in ultra-processed

PBAs. Third, we limited our environmental impact assessment to GHGe, LU, and WU because of sparse data availability on the wider environmental impact of PBAs besides PB meats. For example, we used LCA data for fish fingers as a proxy for PB seafood, as data is not yet available for the growing market of tuna and salmon substitutes. Further, we used data on blue freshwater consumption from agriculture instead of total water use based on available data[47]. Fourth, our price data can be considered conservative as we chose market price values, which include household waste but do not reflect energy prices for preparing the food. This underestimates the costs of cheap commodities such as dry beans where the costs for boiling are not included.

Ultimately, the uptake of the dietary scenarios investigated here will depend on several factors, such as consumer acceptance. We assumed a dietary shift on a population level including wide consumer acceptance and did not stratify the dietary scenarios for different gender, age, or socio-economic groups. The literature on behavioural change, however, suggests that the uptake of plant-based diets is highly dependent on the consumer acceptance of PBAs and WFs, with acceptance and preference varying across population groups[6,48]. Our findings highlight that minimising ASFs consumption has a larger influence on reducing the environmental impact than choosing between PBAs or WFs as replacements. This suggests that the different, more plant-based dietary scenarios that we assessed should be viewed as complementary, targeting different groups that strive to reduce the intake of ASFs. Including PBAs in dietary guidelines, as already prevalent in some countries, can promote the uptake of more sustainable diets based on consumer preferences by presenting a wider choice of plant-based products to select from ref. 49. Importantly, emphasis should thereby be placed on healthier PBAs such as those that avoid ultra-processing and have good amino-acid composition, are low in sodium and provide high nutrient-bioavailability.

Additionally, a key barrier to integrating PBAs into diets is the perception that they are more expensive than ASFs[6,8,50]. By contrast, our estimates of the daily food expenditure revealed that PBA diets can be cost-competitive with the BAU diet in a Swedish context. However, focusing on the cost of diet, as prevalent in the literature[34,51], might provide an incomplete impression of dietary affordability, as socio-economic factors, such as household income and childcare expenditure, impact individual purchasing patterns[51] particularly relevant in a cost of living crisis. As such, future research should expand the affordability assessment to additional variables and stratify for different groups of physical and economic access.

Finally, it is expected that the consumption of PBAs will increase and expand in their product range[15]. Our findings suggest that transitioning to plant-based diets, whether based on PBAs or WFs would lead to substantial reductions in the environmental impacts, while aligning with most nutrition recommendations and being cost-competitive with the current average Swedish diet. More research is therefore imperative to understand the implications of their role in the transformation towards sustainable food systems. We recommend expanding the environmental impact assessment to additional indicators, assessing the bioavailability of nutrients in PBAs when included in whole diets, sociocultural aspects of affordability and acceptability, and conducting the assessment stratified for different socio-economic groups. Hereby future research should continue expanding the assessment to the wide variety of already retail-available PBAs (i.e., seafood, dairy, eggs, cheese, and snacks).

## Methods

In this modelling study, we first derived the current average Swedish diet and composed six plant-based dietary scenarios, where ASFs are either replaced by their respective PBAs (e.g., dairy with PB dairy) or WFs (legumes, grains, vegetables) and then paired the scenarios with respective nutritional, environmental and price data (Supplementary Fig. 1). Our system boundaries were from cradle to consumer-including cooking at the consumer stage if necessary to enable a fair comparison between foods that require different amounts of cooking time (e.g., dried legumes).

### Composing scenarios

As the first step, we derived a current average diet (hereafter BAU) based on current population-scale food consumption provided by the Swedish Statistical Database (Supplementary Methods). The data represents the amount of food available for consumption in Swedish households and institutional kitchens. To reflect the actual consumption of food items and to account for loss at the household stage, we applied food loss and waste ratios provided by FAO[52].

We then composed six scenarios that reflect more plant-based diets and are based on shifting away from the BAU, where ASFs are either replaced by PBAs or WFs (Table 1, Supplementary Methods). These included vegan scenarios where all ASFs are replaced by PBAs (VGNPBA) or WFs (VGNWHOLE); vegetarian scenarios where meat and seafood products are replaced by PBAs (VGTPBA) or WFs (VGTWHOLE), and flexitarian scenarios where 50% of ASFs are replaced with PBAs (FLXPBA) or WFs (FLXWHOLE). For the WFs scenarios, we used the upper reference values for WFs such as legumes proposed by the EAT-Lancet Commission[1]. We constructed the scenarios using two different functional units, mass (grams) and nutritional function (kcals) to address uncertainty in the choice of the functional unit (Supplementary Fig. 2).

### Nutrient analysis

To assess the nutritional quality of the respective diets, we paired each of the included food items with respective data on their nutritional composition. Nutritional values per kg or litre of food product were derived from the Swedish National Food Agency[53]. The database provides information on the nutritional composition of 55 macro-and micronutrients for more than 2000 foods and dishes, including PBAs, available in the Swedish food supply chain. For our analysis, we focused on food products and non-alcoholic beverages and used the median values of the nutritional components for each food item (Supplementary Methods).

Nutrients included in the analysis have been selected in relation to regional nutrition recommendations for healthy and sustainable diets[1,30] and based on their relevance for Swedish conditions following a previous study[54]. To assess the nutritional adequacy of the different diets, we then compared the calculated nutritional content of the different scenarios to daily macro- and micronutrient recommendations provided by the Nordic Nutrition Recommendations for adults[30]. Essential amino acid requirements were from FAO/WHO and amino acid requirements for an adult, reference weight 70 kg from EFSA[55]. We employed the recommendations for moderately active, healthy adults (18–65 years old). Where there are sex differences in recommendations (e.g., for higher daily iron and folate intake in women), we selected the most stringent recommendation. In this way, we ensure that all scenarios meet at least the minimum requirements for all adults regardless of their sex (detailed in Supplementary Data). In the nutritional comparisons, we followed employed methods of nutrition constraints in dietary optimisations of European diets[21,54].

### Environmental impact analysis

We calculated the environmental impact of the dietary scenarios based on available LCA data (Supplementary Methods). We focused on three environmental impact factors that have been used to assess PBAs in comparative LCA studies: greenhouse gas emissions (GHGe) (kg of $CO_2$e), cropland use (LU) (m²), and consumptive freshwater use (WU) (L).

Environmental impact data on food items consumed in Sweden were sourced from Moberg et al.[23]. The study provides the

**Table 1 | Distribution of food categories in the respective dietary scenarios**

| Food Groups | Diet Scenarios | | | | | | |
|---|---|---|---|---|---|---|---|
| | BAU | VGNPBA | VGNWHOLE | VGTPBA | VGTWHOLE | FLXPBA | FLXWHOLE |
| Meats | 190* | 0 | 0 | 0 | 0 | 95 | 95 |
| | 333 | 0 | 0 | 0 | 0 | 166 | 166 |
| Plant-based Meat | 0 | 220 | 0 | 190 | 0 | 110 | 0 |
| | 0 | 375 | 0 | 333 | 0 | 188 | 0 |
| Eggs | 30 | 0 | 0 | 30 | 30 | 15 | 15 |
| | 42 | 0 | 0 | 42 | 42 | 21 | 21 |
| Dairy | 400 | 0 | 0 | 400 | 400 | 200 | 200 |
| | 440 | 0 | 0 | 440 | 440 | 220 | 220 |
| Plant-based Dairy | 0 | 400 | 0 | 0 | 0 | 200 | 0 |
| | 0 | 393 | 0 | 0 | 0 | 196 | 0 |
| Seafood | 30 | 0 | 0 | 0 | 0 | 15 | 15 |
| | 50 | 0 | 0 | 0 | 0 | 25 | 25 |
| Plant-based Seafood | 0 | 30 | 0 | 30 | 0 | 15 | 0 |
| | 0 | 50 | 0 | 50 | 0 | 25 | 0 |
| Vegetables | 165 | 165 | 330 | 165 | 165 | 165 | 250 |
| | 50 | 50 | 100 | 50 | 50 | 50 | 75 |
| Fruits | 150 | 150 | 300 | 150 | 150 | 150 | 230 |
| | 110 | 110 | 220 | 110 | 110 | 110 | 155 |
| Potatoes | 100 | 100 | 100 | 100 | 100 | 100 | 100 |
| | 85 | 85 | 85 | 85 | 85 | 85 | 85 |
| Grains | 170 | 170 | 270 | 170 | 220 | 170 | 220 |
| | 500 | 500 | 550 | 500 | 550 | 500 | 550 |
| Plant fats | 27 | 33 | 33 | 27 | 27 | 27 | 27 |
| | 195 | 240 | 340 | 195 | 195 | 195 | 220 |
| Legumes | 0 | 0 | 150 | 0 | 150 | 0 | 100 |
| | 0 | 0 | 280 | 0 | 280 | 0 | 200 |
| Nuts | 8 | 8 | 50 | 8 | 25 | 8 | 25 |
| | 50 | 50 | 290 | 50 | 150 | 50 | 150 |
| Snacks | 110 | 0 | 0 | 110 | 110 | 55 | 55 |
| | 433 | 0 | 0 | 433 | 433 | 228 | 228 |
| Plant-based Snacks | 0 | 110 | 110 | 0 | 0 | 55 | 55 |
| | 0 | 433 | 433 | 0 | 0 | 205 | 205 |
| Other | 330 | 330 | 330 | 330 | 330 | 330 | 330 |
| | 137 | 137 | 137 | 137 | 137 | 137 | 137 |

Replacements of animal-source foods from the business-as-usual current diet (BAU) with whole foods (VGTWHOLE, VGNWHOLE, FLXWHOLE) or plant-based alternatives (FLXALT, VGTALT, VGNALT) were made on a mass (upper number) and energy basis (lower numbers). BAU Current Average Swedish Diet; VGNPBA Vegan Diet, all ASFs replaced by PBAs; VGTPBA Vegetarian Diet, meat& seafood replaced by respective PBAs; FLXPBA = 50% reduction of ASFs replaced by PBAs; VGNWHOLE=Vegan Diet, all ASFs replaced by WFs; VGTWHOLE= Vegetarian Diet, meat& seafood replaced by WFs; FLXWHOLE = 50% reduction of ASFs replaced by WFs. The category "other" includes sugar and sugar-based products, salt, coffee, soft drinks, and juice. * Upper numbers indicate the mass unit while lower numbers indicate the energy unit in the respective scenario.

environmental impact data for food items consumed in Sweden according to the Swedish Board of Agriculture, hence no data is available on products that have not been included in the database (i.e., PBAs, legumes, soy foods).

Therefore, LCA data for the remaining food items were obtained from additional data sources and harmonised into one inventory database (Supplementary Data). For legumes, soy foods and PBAs we sourced data from a science-based consumer guide for plant-based foods in Sweden[56,57]. For those PBAs, where the report did not provide environmental impact data (i.e., PB cheese, PB seafood and PB snacks), we used data provided by foodDB[58,59].

**Daily food expenditure**
We obtained price values of food products available at Swedish retail chains and calculated median values for the respective food items ($n = 78$) (Supplementary Methods). Food product prices were extracted in June 2022 from the discount retailer *Willys*, which is part of the leading food retail group *Axfood*[60].

**Uncertainty analysis**
In addition to performing each analysis using two functional units (energy and mass) to assess how the choice of functional unit may influence estimated outcomes, we also accounted for uncertainty in the environmental, nutrition, and socio-economic impacts of each food item. For the nutritional analysis, we calculated lower and upper values, 25th and 75th percentile impacts for each macro- and micronutrient from the Swedish National Food Agency database for the respective food items (Supplementary Data). In analysing environmental impacts, we took the 25th and 75th percentile possible impacts based on different commodity supply chains of the PBAs to determine how sensitive food choices are to the environmental impact of the dietary scenarios. Finally, in the daily food expenditure analysis we used the range of

prices of food items extracted from the food retailer to calculate the 25th and 75th percentile values for the more aggregated food groups to provide a price range and better reflect individual consumer behaviour (Supplementary Data).

## Reporting summary
Further information on research design is available in the Nature Portfolio Reporting Summary linked to this article.

## Data availability
The data generated or analysed in this study are provided in the Supplementary Information or Supplementary Data of this manuscript and have been deposited in the Figshare repository database under accession code https://figshare.com/s/cb0d36c368a9121356d3. Source data underlying Figs. 1–3 and Supplementary Figs are available as a Source data file. Source data are provided in this paper.

## Code availability
The code generated to analyse the data is publicly available at https://figshare.com/s/cb0d36c368a9121356d3.

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

## Acknowledgements

We would like to thank Hanna Karlsson Potter for providing stratified data and clarifications on the WWF report. This study was funded in part by the Kamprad Family Foundation for Entrepreneurship, Research and Charity (grant number 20200149) (ACB) and by the Curt Bergfors Foundation (ACB, LG) (FV-2. 1.9-2262-2).

## Author contributions

ACB conceptualised the project with input from all authors. ACB, RM, and MC developed the methodology, conducted the analysis, and carried out the data visualisation. All authors were involved in data interpretation and had full access to the underlying data in the study. LG acquired the funding. ACB wrote the original draft of the manuscript. RM, MC, AW, and LG reviewed and edited the manuscript, and ACB, RM, MC, AW, and LG approved the final manuscript as submitted.

## Funding

## Competing interests

The authors declare no competing interests.
