## [Peer Review File · Nature Communications]

REVIEWER COMMENTS

Reviewer #1 (Remarks to the Author):

First, I would like to congratulate Bunge and colleagues on a great study and high-quality manuscript and I enjoyed reading it.

I had a few discussion points, especially related to the overall message/conclusion of the paper. Very happy for the authors to take these on board or reject them, but I think it might be important to discuss these issues (and I appreciate this discussion might already have taken place in the author team, but it would perhaps then be good to see that reflected in the manuscript):

1. Whilst not exactly phrased as such, the message that media are most probably going to take away from this is that "it does not really matter if you eat fruit and veg or a ultra-processed plant-based burger, either way its better for your health and the planet if you cut meat consumption" - this is perhaps not the message you really want to convey.

The quantification of long term health effects of ultra-processed (plant-based) foods is really still in its infancy. A systematic review is coming out on health impacts of novel plant-based foods later this year, but evidence is thin, whilst there are some interesting twin studies (adjusted for all kinds of macro and micro nutrients) that show rather detrimental effects of UP vs whole food diets (for example one that I recently came across by the Zoe group in the UK - not specifically focused on PB foods though). This makes me a bit anxious about the above message: I appreciate that it would be equally unhelpful in our objective to transform food systems / find evidence that could accelerate behaviour change towards more sustainable and healthier diets to keep burying ourselves into disclaimers on what we don't know yet, but maybe your conclusions are a bit too far at the other end of the spectrum and I would suggest to add some of the above (and below) discussions more prominently in you manuscript.

2. Furthermore, I felt that the fact that the vegetarian and vegan diets do introduce micronutrient deficiencies and aggravates the salt consumption problem is brushed over a bit too fast. The argument that people take supplements anyway is - of course - a valid one, but for example selenium deficiency is a real problem in Sweden, especially among the elderly /vulnerable populations (<https://www.nature.com/articles/ejcn201592>) and these scenarios would potentially increase health inequalities in this way. I think this should be more emphasised: plant-based alternatives are relatively easy to reformulate, so there is likely a solution to the problem here, but its important that audiences/media/policy-makers hear that side of the argument as well! Likewise, the cost differences

are also quickly brushed off as still "cost-competitive". In a cost of living crisis, a few % points increase in HH expenditure on food could form a major problem for some, and hence the alternative protein route does - for now - not seem a viable option for dietary change in the lowest income brackets perhaps? Again, this is perhaps important to mention a bit more upfront.

3. As I mentioned above novel PB foods are relatively easy to reformulate, so mentioning this a bit more prominently might strengthen your argument on the health front (with the caveats mentioned above taken into consideration)

Then, some minor / textual points:

End of methods section 1 – “to account for potential advantages and disadvantages” I wasn’t really sure what that advantages / disadvantages you are referring to here.

Nutritional adequacy results section:

“[...] only VGNPBA and FLXPBA remained adequate intake, due to the fortification of included PBAs seems to miss some words.

There was perhaps some detail lacking on the substitutions in the whole foods scenarios, where the reduction in meat and dairy was ‘absorbed’ by increases in various PB groups including cereals, legumes, vegetables etc. This was all in detail described in appendix 2, but perhaps the headlines could go into the main manuscript. I was also wondering if - rather than using the EAT lancet reference diet for this, it would have made more sense to take either the Swedish guidelines or the actual proportions of substitution food groups, or – in fact – the actual food group ratio of the average Swedish vegan/vegetarian? (Given that you also take the average Swedish [overall] diet). You mention the potential low uptake/consumer acceptance in the discussion, so taking an alternative guidelines might help with that?

Some of the references in the appendix are not correct. I wanted to look at the Moberg et al reference, for example, which is said to be reference 21, but this is not correct. (I think it's now 22 in the main manuscript, but you presumably need it in the list of the references in the appendix as well) I would suggest to check all.

Reviewer #2 (Remarks to the Author):

This study leverages country-specific food consumption estimates and life-cycle assessment data to identify changes in the nutritional adequacy and environmental impacts of Swedish diets when replacing animal-sourced foods with either plant-based alternatives or whole foods across six distinct modelling scenarios as compared to current business-as-usual diets. The analyses and findings present an important contribution to this emerging literature base.

Title: A minor comment - I suggest editing the title to reflect the important and helpful comparisons to whole foods in the analyses, e.g., "Comparing the sustainability benefits of upscaling plant-based alternatives vs. whole foods consumption in Sweden."

Introduction

Line 2 and throughout: "Dietary patterns" in nutrition literature typically refer to a priori indexes or empirically-derived factor scores from food intake data. I would suggest replacing "dietary patterns" with "diets" or "diet scenarios" throughout the whole text (including the abstract, figures, and appendix).

Line 12: I suggest explicitly describing PBAs as processed to provide better rationale for your comparison to "whole" (i.e., minimally processed) foods, i.e., "Contrary to WFs, PBAs are processed foods that aim to mimic..."

Line 31-32: It would benefit this sentence to give a short rationale as to why dietary changes towards sustainability are context-specific.

Methods

Line 54-56: Correct me if something was missed in the Appendix, but I believe the BAU diet was derived assuming all individuals in the Swedish population, including children, consume equal amounts of food. This method would overestimate consumption in children and underestimate consumption in adults. A clarified rationale for and consequences of this analytical decision are needed.

Results

Line 152: I suggest cleaning up formatting for nutrient labels/units in Figure 1. For readability, it may also be worthwhile to be consistent with the shapes denoting the Whole scenarios (e.g., diamonds) vs. the “PBA” scenarios (e.g., squares). Similarly, you can also be consistent with the colors for the VGN, VGT, and FLX scenarios.

Line 182: Similarly, I would suggest cleaning up formatting for environmental impact labels/units.

Discussion

Line 320: Setting consumption data to zero for legumes and soy foods in the BAU diet seems like a major limitation – and likely not reflective of true average dietary intake. It would be helpful to provide some evidence from other studies on Swedish diets to argue for this analytical decision, or else redo all analyses with a more realistic imputed intake value for these two food groups.

Line 348: What would be considered a “healthier” PBAs? Would any data/findings from your analyses shed light on this question?

Response to reviewers 1 and 2 comments on the manuscript

The sustainability benefits of upscaling plant-based alternatives consumption in Sweden

Manuscript ID NCOMMS-23-22747-T

We would like to thank you for the opportunity to revise our manuscript. We kindly thank the two reviewers for their constructive and helpful comments, which have helped us to strengthen our manuscript. Please find our itemized responses to each of the comments below. *Please note, that we have also made our own minor improvements to wording throughout the manuscript.*

Reviewer 1: “First, I would like to congratulate Bunge and colleagues on a great study and high-quality manuscript and I enjoyed reading it. I had a few discussion points, especially related to the overall message/conclusion of the paper. Very happy for the authors to take these on board or reject them, but I think it might be important to discuss these issues (and I appreciate this discussion might already have taken place in the author team, but it would perhaps then be good to see that reflected in the manuscript)”

Reviewer 1: Remarks	Author’s Response
MAJOR	
1. Whilst not exactly phrased as such, the message that media are most probably going to take away from this is that "it does not really matter if you eat fruit and veg or a ultra-processed plant-based burger, either way its better for your health and the planet if you cut meat consumption" - this is perhaps not the message you really want to convey. The quantification of long-term health effects of ultra-processed (plant-based) foods is really still in its infancy. A systematic review is coming out on health impacts of novel plant-based foods later this year, but evidence is thin, whilst there are some interesting twin studies (adjusted for all kinds of macro and micro nutrients) that show rather detrimental effects of UP vs whole food diets (for example one that I recently came across by the Zoe group in the UK - not specifically focused on PB foods though). This makes me a bit anxious about the above message: I appreciate that it would be equally unhelpful in our objective to transform food systems / find evidence that could accelerate behaviour change towards more sustainable and healthier diets to keep burying ourselves into disclaimers on what we don't know yet, but maybe your conclusions are a bit too far at the other end of the spectrum and I	Thank you for pointing attention to this. We have now rephrased several parts of the manuscript to reduce the risk that this would be the message conveyed. Instead, we hope that our main message in this paper is that both diets high in WFs or PBAs can be complementary dietary transition strategies for different consumer groups and that their coexistence is useful for targeting different consumers. However, more research and PBA product improvement are imperative. The changes we have made encompass:  1. Title: Following the valuable suggestion of Reviewer 2, we changed the title to “Sustainability benefits of transitioning from current diets to plant-based alternatives or whole-food diets in Sweden” 2. Abstract Line 33: “Replacing with WFs revealed similar environmental benefits, suggesting that reducing the consumption of ASFs is, from an environmental perspective, more important than the decision to replace them with PBAs or WFs.” - to specify that we

would suggest to add some of the above (and below) discussions more prominently in you manuscript.	only focus on environmental and not nutritional implications here. 3. Based on your suggestions, we have added more nuance throughout the manuscript that there is a need to develop healthier PBAs and that WF scenarios are more affordable than PBA diets (e.g. Line 388-391, 339-341; 411-414; 418-420, 426). Please find detailed information about the changes conducted in our responses hereafter.
2. Furthermore, I felt that the fact that the vegetarian and vegan diets do introduce micronutrient deficiencies and aggravates the salt consumption problem is brushed over a bit too fast. The argument that people take supplements anyway is - of course - a valid one, but for example selenium deficiency is a real problem in Sweden, especially among the elderly /vulnerable populations (https://www.nature.com/articles/ejcn201592) and these scenarios would potentially increase health inequalities in this way. I think this should be more emphasised: plant-based alternatives are relatively easy to reformulate, so there is likely a solution to the problem here, but its important that audiences/media/policy-makers hear that side of the argument as well! Likewise, the cost differences are also quickly brushed off as still "cost-competitive". In a cost of living crisis, a few % points increase in HH expenditure on food could form a major problem for some, and hence the alternative protein route does - for now - not seem a viable option for dietary change in the lowest income brackets perhaps? Again, this is perhaps important to mention a bit more upfront.	Thank you for raising this important concern and for drawing attention to the study on the selenium intake. We have now amended several parts of the manuscript to reflect on micronutrient deficiencies, those encompass:  1. We added the sentence in lines 391-394: "However, PBAs as processed foods can be reformulated to improve their healthiness, with particular emphasis should be on reducing sodium content, which our findings revealed is currently too high in PBAs, and promoting nutrients of concern in the Swedish population." 2. We added the sentence in lines 397-399: ". Contrary, ultra-processing has been linked to various adverse long-term health outcomes, highlighting the necessity for further research into the health implications of diets rich in ultra-processed PBAs" 3. Line 311-313: " The sodium content increased in the PBA scenarios compared to the BAU (11-24%) adding to the concern that PBAs have too high levels of added salt while it decreased in the WF scenarios. Too high sodium intake is a major Public health concern" 4. We added the sentence in Line 320: "Low selenium intake was revealed in all

	scenarios, including the BAU, which reflects existing evidence that selenium intake is a concern in Sweden. And have cited the suggested paper by Alehagen et al. Regarding cost-competitiveness, this is an important point and we now have revised our wording across the whole manuscript. These changes encompass:  1. Abstract: "Replacing ASFs with PBAs can reduce the environmental impact of current Swedish diets while being cost-competitive and meeting most nutritional recommendations, but slightly increases food expenditure." 2. Line 337-340: "Economically, we estimated that all assessed scenarios were similar cost-competitive to the BAU diet when comparing median prices.... These results suggest that healthier and more environmentally sustainable diets can be obtained without much alteration of food expenditure costs in Sweden, depending on which end of the price range consumers make their purchases." 3. Line 261: We replaced "slightly more expensive" with "priced higher than" 4. Line 425: "Our estimates of the daily food expenditure revealed that PBA diets can be cost-competitive (this is followed by a longer discussion on all the external factors that impact purchasing power where we now have also added "particularly relevant in a cost of living crisis" in line 369)
3. As I mentioned above novel PB foods are relatively easy to reformulate, so mentioning this a bit more prominently might strengthen your argument on the health front (with the caveats mentioned above taken into consideration)	Thank you for raising this very important point that we've been well aware of and have extensively discussed within the author group. In our initial submission, we placed particular emphasis on it in our discussion, lines 325-329:

	“...ongoing research focuses on developing next-generation PBAs that will be healthier, tastier, more sustainable and produced locally from Swedish raw materials. For example, by using fermentation practices to improve the bioavailability of nutrients in PBAs...” To ensure more emphasis on the reformulation potential in already retail-available PBAs, we have now added a sentence in the discussion section (line 391) to strengthen this argument. The sentence reads: “However, it is important to note that PBAs as processed foods can be reformulated to improve their healthiness, with particular focus on reducing the sodium content which we revealed is currently too high in PBAs and promoting nutrients of concern in the Swedish population.” And added the following reference: McClements, D. J. Ultraprocessed plant-based foods: Designing the next generation of healthy and sustainable alternatives to animal-based foods. Compr. Rev. Food Sci. Food Saf. 22, 3531–3559 (2023).
MINOR	
End of methods section 1 – “to account for potential advantages and disadvantages” I wasn’t really sure what that advantages / disadvantages you are referring to here.	Thank you for pointing our attention to this. Here, we aimed to refer to the fact that choosing the functional unit can have a decisive impact on the environmental, nutritional and economic performance of food items. For example, assessing the environmental impact on a mass basis comparison favours energy-dense foods and disadvantages foods with high water content such as vegetables. We provide a detailed explanation of this in our supplementary material. Consequently, we have now decided to remove “advantages and disadvantages” from the main manuscript and refer to the supplementary material instead, where we have added more nuance to our explanation.
Nutritional adequacy results section: “[...] only VGNPBA and FLXPBA remained adequate intake, due to the fortification of included PBAs seems to miss some words.	Thank you, we have now revised this sentence to enhance clarity. It now reads as: Line 200: “Vitamin D content decreased in all scenarios and only VGNPBA and FLXPBA remained adequate intake, which is

	attributable to the Vitamin D fortification of included PBAs.”
There was perhaps some detail lacking on the substitutions in the whole foods scenarios, where the reduction in meat and dairy was ‘absorbed’ by increases in various PB groups including cereals, legumes, vegetables etc. This was all in detail described in appendix 2, but perhaps the headlines could go into the main manuscript. I was also wondering if - rather than using the EAT lancet reference diet for this, it would have made more sense to take either the Swedish guidelines or the actual proportions of substitution food groups, or – in fact – the actual food group ratio of the average Swedish vegan/vegetarian? (Given that you also take the average Swedish [overall] diet). You mention the potential low uptake/consumer acceptance in the discussion, so taking an alternative guidelines might help with that?	Thank you for drawing our attention to this. We have now added additional and more detailed information on the composition of the whole foods scenario in the main methods section (line 101 ff.). The sentence now reads as: “...where ASFs are either replaced by their respective PBAs or WFs (legumes, grains, vegetables) and then paired the scenarios...” and hope that this provides the readers with a better understanding of the composition of whole food scenarios.” We further added the sentence in line 117-118: “For the WFs scenarios, we used the reference values for vegetables, fruits, legumes, nuts, and grains proposed by the EAT-Lancet Commission” Regarding constructing the scenarios using the EAT-Lancet reference diet. Please see below the several reasons for our decision to construct the scenarios using EAT-Lancet. These informations have now been added to our supplementary material (page 3) titled “Methodological validation”. 1) The Swedish guidelines only provide recommendations with specific intake values for selected food groups but not for all, such as legumes which we aimed to have included in our scenarios. 2) Taking the actual food group ratio of the average Swedish vegetarian/vegan is a great suggestion, but unfortunately, we are unaware of the existence of such data and our renewed targeted search has not yielded any results in this regard. 3) The EAT-Lancet reference diet provides specific numerical guidelines for legume intake in environmentally sound and healthy diets. Their recommendations for legume intake are based on extensive literature reviews.

	For those reasons, we wish to continue with using the EAT-Lancet reference diet as it further enables comparing our study with the emerging scientific evidence base that investigates more healthy and sustainable dietary scenarios.
Some of the references in the appendix are not correct. I wanted to look at the Moberg et al reference, for example, which is said to be reference 21, but this is not correct. (I think its now 22 in the main manuscript, but you presumably need it in the list of the references in the appendix as well) I would suggest to check all.	Thank you for this attentive information. We have now checked all the references again to ensure that they are correct and consistent and have added the Moberg et al reference to the supplemental material.

Reviewer 2: “This study leverages country-specific food consumption estimates and life-cycle assessment data to identify changes in the nutritional adequacy and environmental impacts of Swedish diets when replacing animal-sourced foods with either plant-based alternatives or whole foods across six distinct modelling scenarios as compared to current business-as-usual diets. The analyses and findings present an important contribution to this emerging literature base. “

Reviewer 2: Remarks	Author’s Response
Title	
Title: A minor comment - I suggest editing the title to reflect the important and helpful comparisons to whole foods in the analyses, e.g., “Comparing the sustainability benefits of upscaling plant-based alternatives vs. whole foods consumption in Sweden.”	Thank you for the suggestion. We have now rephrased the title, which now reads as: “Sustainability benefits of transitioning from current diets to plant-based alternatives or whole-food diets in Sweden”
INTRODUCTION	
Line 2 and throughout: “Dietary patterns” in nutrition literature typically refer to a priori indexes or empirically-derived factor scores from food intake data. I would suggest replacing “dietary patterns” with “diets” or “diet scenarios” throughout	Thank you for bringing this to our attention. We have now exchanged “dietary pattern” with “diet”, “dietary scenario” or “dietary behaviour” depending on which term was more appropriate in the respective context. Changes have been conducted both in the main text as well as in the appendix.

the whole text (including the abstract, figures, and appendix).	
Line 12: I suggest explicitly describing PBAs as processed to provide better rationale for your comparison to “whole” (i.e., minimally processed) foods, i.e., “Contrary to WFs, PBAs are processed foods that aim to mimic...”	Thank you for drawing our attention to this. We have now amended the sentence in line 63 according to your suggestion. It now reads as “Contrary to WFs, PBAs are processed foods that aim to mimic the structure, texture, and sensorial properties of the ASF they intend to replace”.
Line 31-32: It would benefit this sentence to give a short rationale as to why dietary changes towards sustainability are context-specific.	Thank you for the suggestion. This sentence has now been revised in line with the suggestion and now reads as: Line 82: “As pathways for more sustainable diets are highly context-specific- depending on the national burden of diseases, environmental challenges related to respective food systems and cultural traditions”- such studies would benefit from being situated in specific cultural contexts.” To strengthen that argument, we now cite the following paper: “Biesbroek, S. et al. (2023) ‘Toward healthy and sustainable diets for the 21st century: Importance of sociocultural and economic considerations’, Proceedings of the National Academy of Sciences, 120(26), p. e2219272120. Available at: https://doi.org/10.1073/pnas.2219272120.” which provides a detailed discussion on why sustainable diets depend on the national context and why this should be reflected in strategies for transforming diets. The paper was published after our initial submission, which is why we could not quote it there.
METHODS	
Line 54-56: Correct me if something was missed in the Appendix, but I believe the BAU diet was derived assuming all individuals in the Swedish population, including children, consume equal amounts of food. This method would overestimate consumption in children and underestimate consumption in adults. A	Thank you for raising this important point. We agree that taking the average consumption across age groups might be a limitation. We now provide a clarified detailed rationale for this analytical decision in our supplemental material on page 3 and refer to it in the main manuscript. This includes an acknowledgement that this is a simplification that might under- or overestimate specific population groups, but that it is standard practice to adapt dietary guidelines to different population groups. Please see below for a more detailed response on why this approach was taken. 1)As an average kcal intake we retrieved 2450 kcal for the BAU diet. For comparison, the EAT-Lancet study uses a reference intake of 2500 kcal/day for a healthy diet—they state that consuming 2500 kcal per day corresponds to the average energy needs of a 70-kg man aged 30

clarified rationale for and consequences of this analytical decision are needed.	years and a 60-kg woman aged 30 years whose level of physical activity is moderate to high. This reference level was also used in studies in the Swedish contexts (e.g.: https://www.sciencedirect.com.ezp.sub.su.se/science/article/pii/S0002916522003677) 2) According to the Swedish National Food Agency, 2500 kcal is the upper recommendation for younger women and the lower to median recommendation for men, depending on their age. As such, our derived reference diet reflects the recommended intake for some adults, and even presents an overestimation for other adult groups. https://www.livsmedelsverket.se/livsmedel-och-innehall/naringsamne/energi-kalorier 3) While it is an average for the whole society, Jordbruksverket (The Swedish agency) does not provide stratified data for different age or sex groups. In addition, we now state that: “We recognise that, even if we follow standard practices to adapt dietary guidelines to different population groups, this is a simplification that particularly overestimate consumption for children and elderly, while potentially underestimating consumption for very physically active adults.”
---	---

RESULTS

Line 152: I suggest cleaning up formatting for nutrient labels/units in Figure 1. For readability, it may also be worthwhile to be consistent with the shapes denoting the Whole scenarios (e.g., diamonds) vs. the “PBA” scenarios (e.g., squares). Similarly, you can also be consistent with the colors for the VGN, VGT, and FLX scenarios.	Thank you for the valuable suggestion. The figure has now been amended in line with your suggestions.
Line 182: Similarly, I would suggest cleaning up formatting for environmental impact labels/units.	Thank you here as well, we have now formatted the figure in line with your suggestions. In addition, we also formatted Figure 3.

DISCUSSION

Line 320: Setting consumption data to zero for legumes and soy	This is an important point that has been extensively discussed within the author group. While we acknowledge that excluding legumes in the BAU diet is a limitation, it reflects the current deficits of available
---	---

foods in the BAU diet seems like a major limitation – and likely not reflective of true average dietary intake. It would be helpful to provide some evidence from other studies on Swedish diets to argue for this analytical decision, or else redo all analyses with a more realistic imputed intake value for these two food groups.	data on legume intake in Sweden (please see lines 383-387 in the main manuscript and page 3 in the supplemental material where have now added additional information). As we stated in the discussion and more detailed in the appendix, setting the consumption to zero was decided because consumption data of legumes was not provided. The latest data on legume consumption in Sweden is from the national study Riksmaten corresponding to consumption in the year 2010/2011. The data suggests that the daily per capita consumption of legumes in Sweden was 12 g, but eating patterns differ considerably between individuals and only 50% of Swedish women and 44% of Swedish men included legumes in their diet (NFA, 2012). This information has now been added to the appendix, please see page 3. As such, it can be assumed that a BAU diet with zero consumption of legumes reflects the dietary behaviour of several Swedish people. On the other hand, we understand that the BAU diet we are using here does not reflect consumers who have already integrated legumes into their diet. To reflect that, we now added a sentence in the discussion which reads as: Lines 385-387 “Hence and in line with the available dietary intake data, we set the consumption data of PBAs, legumes, and soy foods to zero in the BAU diet, which does not reflect consumers already including PBAs or legumes in their diet.”
Line 348: What would be considered a “healthier” PBAs? Would any data/findings from your analyses shed light on this question?	Thank you for drawing our attention to this. We have now added more detailed information and the rephrased sentence now reads as Line 421 ff: “Importantly, emphasis should thereby be placed on healthier PBAs such as those that avoid ultra-processing and that have good amino-acid composition, are low in sodium and provide high nutrient- bioavailability.”

REVIEWER COMMENTS

Reviewer #1 (Remarks to the Author):

The authors have addressed all my concerns. I would urge them to take good care of the press release to make sure the message they want to convey will also be the one picked up by the media.

Reviewer #2 (Remarks to the Author):

Immense gratitude to the authors for diligently incorporating the reviewers' feedback into the updated manuscript.

A small last note - revisiting the limitations of failing to include legume intake in the BAU diet (in line 387 of the tracked manuscript), I would suggest adding the authors' postulated expectations on how results might be different if you were able to include legume / soy intake. How would the nutritional adequacy results change? The environmental impact results? The food expenditure results? It can be broad strokes here (and a nod to the need for more research to confirm your results).

Response to reviewers 1 and 2 remarks

Sustainability benefits of transitioning from current diets to plant-based alternatives or whole-food diets in Sweden

Manuscript ID NCOMMS-23-22747A

Reviewer1:

The authors have addressed all my concerns. I would urge them to take good care of the press release to make sure the message they want to convey will also be the one picked up by the media.

Authors Response: We would like to thank the anonymous reviewer again for having engaged with our manuscript in such detail and for providing valuable input that improved not only the quality of our work but also helped us to improve our wording to convey the intended message. We will ensure to handle media engagements with care.

Reviewer 2: Immense gratitude to the authors for diligently incorporating the reviewers' feedback into the updated manuscript.

A small last note - revisiting the limitations of failing to include legume intake in the BAU diet (in line 387 of the tracked manuscript), I would suggest adding the authors' postulated expectations on how results might be different if you were able to include legume/soy intake. How would the nutritional adequacy results change? The environmental impact results? The food expenditure results? It can be broad strokes here (and a nod to the need for more research to confirm your results).

Author's Response: We would like to thank the anonymous reviewer for their earlier diligent engagement with the manuscript, which we have greatly benefitted from.

Regarding the recommendation to add more information on how the environmental, nutritional and economic performance would have potentially changed if the data availability had enabled to include legumes, we have now added the following sentence to the discussion: *"Including legumes in the BAU diet would have slightly altered the environmental, economic and nutritional performance, such as a potential increase in fiber intake"*.